# DNA Methylation Signatures Predict Cytogenetic Subtype and Outcome in Pediatric Acute Myeloid Leukemia (AML)

**DOI:** 10.3390/genes12060895

**Published:** 2021-06-10

**Authors:** Olga Krali, Josefine Palle, Christofer L. Bäcklin, Jonas Abrahamsson, Ulrika Norén-Nyström, Henrik Hasle, Kirsi Jahnukainen, Ólafur Gísli Jónsson, Randi Hovland, Birgitte Lausen, Rolf Larsson, Lars Palmqvist, Anna Staffas, Bernward Zeller, Jessica Nordlund

**Affiliations:** 1Department of Medical Sciences, Molecular Precision Medicine and Science for Life Laboratory, Uppsala University, 752 37 Uppsala, Sweden; Olga.Krali@medsci.uu.se; 2Department of Women’s and Children’s Health, Uppsala University, 752 37 Uppsala, Sweden; 3Department of Medical Sciences, Cancer Pharmacology and Computational Medicine, Uppsala University, 751 85 Uppsala, Sweden; christofer.backlin@gmail.com (C.L.B.); rolf.larsson@medsci.uu.se (R.L.); 4Department of Pediatrics, Queen Silvia Children’s Hospital, 416 85 Gothenburg, Sweden; vobjab@gmail.com; 5Department of Clinical Sciences, Pediatrics, Umeå University Hospital, 901 85 Umeå, Sweden; ulrika.noren-nystrom@umu.se; 6Department of Pediatrics, Aarhus University Hospital, DK-8200 Aarhus, Denmark; hasle@dadlnet.dk; 7Children’s Hospital, Helsinki University Central Hospital, Helsinki, and University of Helsinki, 00290 Helsinki, Finland; kirsi.jahnukainen@ki.se; 8Department of Pediatrics, Landspitali University Hospital, 101 Reykjavík, Iceland; olafurgi@landspitali.is; 9Center of Medical Genetics and Molecular Medicine, Haukeland University Hospital, 5009 Bergen, Norway; randi.hovland@helse-bergen.no; 10Department of Pediatrics and Adolescent Medicine, Rigshospitalet, University of Copenhagen, 2100 Copenhagen, Denmark; birgitte.lausen@rh.regionh.dk; 11Department of Clinical Chemistry and Transfusion Medicine, University of Gothenburg, 41346 Gothenburg, Sweden; lars.palmqvist@clinchem.gu.se (L.P.); anna.staffas@clinchem.gu.se (A.S.); 12Division of Paediatric and Adolescent Medicine, Oslo University Hospital, 0450 Oslo, Norway; bzeller@ous-hf.no

**Keywords:** DNA methylation, pediatric AML, acute myeloid leukemia, classification, 450k array, epigenetics, subtyping

## Abstract

Pediatric acute myeloid leukemia (AML) is a heterogeneous disease composed of clinically relevant subtypes defined by recurrent cytogenetic aberrations. The majority of the aberrations used in risk grouping for treatment decisions are extensively studied, but still a large proportion of pediatric AML patients remain cytogenetically undefined and would therefore benefit from additional molecular investigation. As aberrant epigenetic regulation has been widely observed during leukemogenesis, we hypothesized that DNA methylation signatures could be used to predict molecular subtypes and identify signatures with prognostic impact in AML. To study genome-wide DNA methylation, we analyzed 123 diagnostic and 19 relapse AML samples on Illumina 450k DNA methylation arrays. We designed and validated DNA methylation-based classifiers for AML cytogenetic subtype, resulting in an overall test accuracy of 91%. Furthermore, we identified methylation signatures associated with outcome in t(8;21)/*RUNX1-RUNX1T1*, normal karyotype, and *MLL/KMT2A*-rearranged subgroups (*p* < 0.01). Overall, these results further underscore the clinical value of DNA methylation analysis in AML.

## 1. Introduction

Acute myeloid leukemia (AML) in children is a heterogeneous disease with variable clinical outcome. In pediatric as well as adult AML, different prognostic subtypes are characterized either by recurrent cytogenetic aberrations or a cytogenetically normal karyotype (NK) [1]. However, as many as 44% of pediatric AML cases remain cytogenetically unclassified, resulting in a large highly heterogeneous group with not otherwise specified (NOS) cytogenetic classification [2]. Not only does molecular subtype provide prognostic information, but it is important prior knowledge for targeted assays that measure the response to induction treatment measured by minimal residual disease [3,4]. In addition, cytogenetic analysis not only yields important information on diagnosis, prognosis, and follow-up, it provides information about the initiating events in the process of leukemogenesis [5]. Furthermore, NK as determined by conventional cytogenetic analysis may conceal diagnostically relevant rearrangements [6]. Accurate subtyping is therefore exceedingly important for diagnostics as well as future exploratory research [7].

In addition to cytogenetic aberrations, DNA methylation may reflect underlying biological processes that are relevant for studying clinical outcomes. Aberrant DNA methylation has been widely observed in correlation with cytogenetic subtypes of AML, however most studies have been performed predominantly in adults [8,9,10,11,12,13], where the molecular landscapes and clinical outcomes are different from those observed in children [14,15]. However, the strong evidence for a correlation between epigenetic factors, molecular cytogenetic subtype, and outcome in adult AML renders DNA methylation an interesting molecular marker for prognostication also in pediatric AML. A handful of studies have examined DNA methylation in depth in pediatric AML in general [16,17] as well as within MLL/*KMT2A*-rearranged [18], inv(16)/*CBFB-MYH11* [19], and t(8;21)*/RUNX1-RUNX1T1* [20,21] subtypes. Several other studies have identified putative signatures predictive of outcome across subtypes [22], as well as within specific AML subtypes [19,21]. Taken together, these studies provide compelling evidence that DNA methylation captures information about underlying the various biological processes taking place within this heterogeneous disease.

In the present study, we examined 125 Nordic pediatric patients with AML genome-wide DNA methylation analysis and integrated the DNA methylation profiles with clinical data in order to investigate whether variable DNA methylation patterns hold diagnostic or prognostic information in pediatric AML.

## 2. Materials and Methods

### 2.1. DNA Samples

Bone marrow (BM) or peripheral blood (PB) from 142 samples taken at diagnosis (*n* = 123) or relapse (*n* = 19) collected from 125 unique AML patients diagnosed at Nordic Pediatric Centers between 1997–2008 were analyzed in the study (Appendix A). The patients were treated according to NOPHO93 and NOPHO2004 protocols [23,24]. Diagnosis was determined by utilizing BM aspirates for morphology and immunophenotype, while G-band karyotyping, fluorescence in situ hybridization (FISH), and reverse transcription polymerase chain reaction (RT-PCR) was used for cytogenetic characterization. Leukemic cells isolated from AML patient samples were centrifuged by Ficoll-isopaque (Pharmacia, Uppsala, Sweden), pelleted and frozen in liquid nitrogen. All samples selected for the study contained ≥70% blasts as evaluated by microscopy. DNA was extracted from pelleted cells with the AllPrep DNA/RNA Mini Kit according to manufacturer’s instructions (Qiagen, Hilden, Germany).

### 2.2. DNA Methylation Assay

First, 500 ng of genomic DNA was treated with sodium bisulfite according to the manufacturer’s specifications (EZ DNA methylation Gold, Zymo Research, Irvine, CA, USA). Then, 200 ng of the converted DNA was loaded on an Infinium HumanMethylation 450k BeadChip Assay and processed according to the manufacturer’s instructions (Illumina, San Diego, CA, USA).

### 2.3. Data Preprocessing

Data analysis was performed in Python [25], using SciPy [26] in the Jupyter Notebook environment [27]. All scripts are available on GitHub (https://github.com/Molmed/Krali-Palle_2021, accessed on 4 June 2021). The raw IDAT files were processed using methylprep (https://pypi.org/project/methylprep/, accessed on 13 January 2021) to produce a *β*-value matrix, where *β* = 0 corresponds to no methylation, *β* = 1 complete methylation of a given CpG site. *β*-values that failed to pass the *p*-value cutoff (<0.05) were replaced with NaNs. Normal-exponential convolution using out-of-band probes used for background correction was used to normalize the *β*-value distribution of Infinium type I and II probes. Probes were filtered using an empirical method previously described [28], and further filtered to remove CpG sites with >10% missing values, resulting in a total of 406,830 autosomal CpG sites for downstream analyses. Multidimensional scaling (MDS) was performed for outlier detection resulting in high quality data from 142 DNA samples for analysis. No batch effects were identified among the different batches, while running batch effect correction with Combat [29], thus the uncorrected data were used.

### 2.4. Subtype Prediction with DNA Methylation

Subtype classification was performed using diagnostic samples from 99 patients with the following cytogenetic subtypes: t(8;21)/*RUNX1-RUNX1T1*, inv(16)/*CBFB-MYH11*, t(15;17)*PML-RARA*, *MLL*/*KMT2A*-rearranged, NK, mono 7, sole +8, and 3q21q26. Samples with 11q23/*MLL*/*KMT2A*, t(9;11), t(10;11), t(11;19) were merged into one *MLL/KMT2A*-rearranged group. The samples with undefined cytogenetic subtype (no result and other clonal abnormalities) were excluded from the classifier design. The dataset was split into a training set used to train the classifiers (66%, N = 66) and a test set used to evaluate the classifier performance (33%, N = 33). For DNA methylation classification, we tested eight different commonly used machine learning algorithms with Scikit-learn [30] and one neural network with Keras (https://github.com/fchollet/keras, accessed on 20 February 2021). Nested k-fold cross-validation was used to identify the best hyperparameters per classification model (Appendix A) and to select the best model in terms of accuracy score on the validation dataset. During k-fold cross-validation, k-1 parts were used as a training set and the remaining part as a validation set. Each classifier was trained using an outer 5-fold cross-validation to obtain the mean accuracy per optimized classifier and 3-fold cross validation was run in the inner loop to identify the best set of hyperparameters per classifier. In each iteration, any missing DNA methylation values were imputed by using the median. A one-vs.-rest multiclass strategy was implemented where each class (subtype) was trained against all the other classes. A confusion matrix and a classification report with precision, recall, and f1-score were generated from the predictions.

Feature selection (FS) was performed prior to modeling to capture the CpG sites that represent the most variability in the dataset using the combination of two unsupervised approaches. For FS, a 5-fold cross-validation approach was followed and all of the selected CpGs per fold from both methods were retained. First, a PCA-approach was implemented where a maximum of 20 most informative CpG sites (per fold) per component from the first 15 principal components were selected, resulting in 955 CpG sites. In a second approach, CpG sites with low variance and high correlation (LVHC) were removed. LVHC thresholds of <0.1 data variance and >70% correlation were set, resulting in 456 CpG sites. The union of the two FS approaches resulted in 1300 CpG sites for downstream analysis.

The Wilcoxon signed-rank test was used to compare the performance of the models. Finally, a permutation test (*n* = 1000 permutations) was performed to validate that the model performed well due to the feature and target dependency and not due to stochastic noise. To identify differentially methylated CpG sites in the different subtypes, Mann–Whitney U tests were applied. Benjamini–Hochberg multiple testing correction was applied to *p*-values.

### 2.5. Survival Analysis

Patients were grouped within subtype based on hierarchical clustering of selected CpG sites. Relapse free survival (RFS) and overall survival (OS) where event was defined as death from any cause was used to evaluate differences in outcome. Survival plots were generated with the Kaplan–Meier method (https://github.com/erdogant/kaplanmeier, accessed on 19 March 2021). The log-rank test and Cox regression analysis were used to analyze the differences between DNA methylation groups with respect to RFS and OS.

### 2.6. Data Visualization

Matplotlib [31] and Seaborn [32] were used for all visualizations. Heatmaps with annotation bars ordered by hierarchical clustering were created by a custom function AnnotHeatMap (https://github.com/Molmed/Krali-Palle_2021, accessed on 4 June 2021).

## 3. Results

### 3.1. Overview of the DNA Methylome in Pediatric AML

A total of 142 DNA samples from 125 unique pediatric AML patients were analyzed on Illumina 450k DNA methylation arrays (Appendix A). The samples included 123 DNA samples taken at AML diagnosis and 19 samples taken at relapse. Table 1 summarizes the cytogenetic subtypes and French-American-British (FAB) classification of the AML patients included in the study. To obtain an initial view of the variation in CpG site methylation in our dataset, we subjected the complete set of methylation data (142 samples across 406,830 autosomal CpG sites) in 2D space with the help of Uniform Manifold Approximation and Projection for Dimension Reduction (UMAP) [33], which resulted in clustering of samples by cytogenetic subtype (Appendix A). A heatmap and hierarchical clustering of all samples using the 10,000 most variable CpG sites are shown in Appendix A. Because the samples with any t(11q23) clustered together, we merged them together in one *MLL/KMT2A*-rearranged subgroup for subsequent analyses.

After performing FS, the remaining 1300 CpG sites resulted in clearly defined clusters separating the known AML subtypes, which suggests that our FS strategy retained significant information while reducing noise (Figure 1). As expected, patients belonging to subtypes defined by *MLL/KMT2A*-rearrangments, t(8;21)*RUNX1/RUNX1T1*, inv(16)*CBFB/MYH11*, mono 7, and t(15;17)*PML-RARA* clustered by their known cytogenetic subtype, indicating a clear relationship between cytogenetic subgroup and DNA methylation. The patients with cytogenetic “normal karyotype” (NK) formed two loosely defined clusters. The large proportion of the cytogenetically undefined samples (no result from diagnostic cytogenetic workup) in this cohort demonstrate DNA methylation similarities with the known subtypes, and not of new independent subtypes.

### 3.2. Prediction of Cytogenetic Subtype with DNA Methylation

To address subtype membership of the cytogenetically undefined samples in our cohort, we built cytogenetic subtype classifiers for eight of the known cytogenetic subtypes present in our cohort using diagnostic AML samples. In total, nine different classification methods were tested and the best performing model was the neural network with mean validation accuracy of 85%, followed by nearest centroid (82%) and random forest (80%) (Appendix A). As the neural network model performed best in terms of accuracy, we proceeded to fit the classifier on all 66 training samples and tested the performance on the remaining 33 unseen patient samples (test set). Finally, the classifier was fitted on all 99 samples of known subtype and used to predict the cytogenetic subtype of the 43 unknown samples. The classifier predicted the correct subtype in 91% (30/33) of test samples, indicating that cytogenetic subtype is highly correlated with DNA methylation levels (permuted *p*-value = 0.001; Appendix A). Our resulting DNA methylation classifier correctly predicted samples of cytogenetic subtypes *MLL/KMT2A*-rearranged, inv(16)/*CBFB-MYH11*, and t(8;21)/*RUNX1-RUNX1T1* with the highest precision and recall scores, followed by NK (Figure 2a; Table 2). The small number of samples (*n* ≤ 5) precludes further evaluation of the t(15;17)/*PML-RARA*, sole +8, mono 7, and 3q21q26 classifiers, although our data provide evidence that these subtypes can be captured by DNA methylation given enough data for training (Table 2). To provide additional validation for our subtype classifier, we predicted the subtype of 14 of the matched samples taken at AML relapse where the diagnostic sample was *MLL/KMT2A*-rearranged, t(8;21)/*RUNX1-RUNX1T1*, NK or inv(16)/*CBFB-MYH11*. The relapse samples are expected to maintain the same subtype as at diagnosis, and were not included in the classifier design (Appendix A). In total, 11 of the 14 predicted subtypes matched with the diagnostic subtype, adding an extra layer of validation to evaluate the predictive performance of the classifier (Appendix A).

The cytogenetic subtypes of the remaining 29 cytogenetically undefined diagnostic samples were predicted by the classifier. In total, 18 were predicted as NK, seven as *MLL*-rearranged, two as t(8;21)/*RUNX1-RUNX1T1*, and two as mono7. The methylation profiles of the newly classified samples closely matched those of the group of original samples used to design the classifier and are referred to as “subtype-suspected”. The heatmap and hierarchical clustering of 127 diagnostic and samples taken at relapse for both characterized and predicted subtypes belonging to the four groups show that most of the data are clustered by subtype driven by differential DNA methylation levels for the 1300 selected CpGs (Figure 2b).

For the most part, the samples and classes performed neat homogenous clusters based on the DNA methylation levels. One notable deviation is the split of the NK class into two clusters (Figure 2b). Additionally, the three cytogenetically defined *MLL*-rearranged samples (AML002, AML004, AML086) that deviated in the UMAP clustering, also cluster together with the large green cluster of NK samples and six more including the fourth AML sample that deviated (AML015, AML33_r, AML041, AML044_r, AML097, and AML099) on the other NK smaller cluster (orange). The NK samples (AML021, AML049, and AML068) cluster together with the inv(16)/*CBFB-MYH11* samples. The high subtype probability scores from the classifier (Appendix A) strongly indicate that at least based on DNA methylation evidence, that these samples either were incorrectly classified at AML diagnosis or that these patients have other underlying molecular events driving their deviating DNA methylation profile.

### 3.3. Intra-Subtype Heterogenetity in NK-AML

As observed in Figure 1 and Figure 2b, the NK group split into two distinctive groups based on DNA methylation levels. To further investigate intra-group heterogeneity of NK-AML, we performed unsupervised hierarchical clustering of all of the diagnostically defined NK samples (N = 30) as well as DNA methylation predicted NK samples (N = 15) using the previously defined 1300 CpG sites. Two well-defined clusters denoted as Cluster A and Cluster B emerged (Figure 3). We compared key molecular and clinical features available for this dataset, such as mutational status of genes of clinical interest (*CEBPA*, *WT1*, *FLT3*, and *NPM1*) from previous work [34] as well as features such as age, sex, and clinical outcome between the two cluster groups (Appendix A). Collectively, *CEBPA*, *FLT3*, *NMP1*, and *WT1* mutations were enriched in Cluster B, with significantly higher proportion of patients in Cluster B (68%) displaying one or more mutations in comparison to only 18% of the A group (Fisher’s exact *p*-value 0.002). Although no significant difference was observed when each gene was analyzed independently (Appendix A), most *CEBPA* mutated patients clustered together in a sub-cluster of Cluster B, suggesting a common underlying epigenetic signature associated with this mutation. Of the clinical features analyzed, the patients in Cluster B were significantly older at AML diagnosis (mean 10.8 years) than those in Cluster A (mean 4.5 years, *t*-test *p*-value 0.00004).

### 3.4. Cytogenetic-Specific DNA Methylation Signatures

The CpG sites important for each cytogenetic subtype were determined by analyzing the 1300 CpG sites in a one-vs.-rest approach. Information about the CpG sites (adjusted *p*-value < 0.05) that separated each subtype from all the rest and the CpG sites that were unique to each subtype are presented in Table 3. More detailed information about the 105 unique CpG sites obtained after CpG annotation to the human genome can be found in Appendix A. We were unable to detect subtype-specific DNA methylation for the Sole + 8 and 3q21q26 subtypes due to low numbers of samples. Amongst the genes identified, include those genes previously known to be differentially expressed or methylated in a subtype-specific pattern such *ASB2* [35] and *PLAUR* [36] in *MLL/KMT2A*-rearranged and *DPF3* in inv(16)/*CBFB-MYH11* [17].

### 3.5. Putatively Prognostic DNA Methylation Signatures

Next, we investigated if the 1300 CpGs hold prognostic information. To avoid confounding results due to the subtype-specific differences in this dataset, we performed survival analysis within each of the t(8;21)/*RUNX1-RUNX1T1*, *MLL/KMT2A*-rearranged, and NK subtypes, where there was an adequate number of patients for analysis. First, we analyzed the 1300 CpG sites for differences between patients with t(8;21)/*RUNX1-RUNX1T1* (N = 18), *MLL/KMT2A*-rearranged (N = 19) or NK (N = 26) subtypes who experienced a relapse in comparison to those who did not (top 50 CpG sites relapse vs. no relapse; *p*-value < 0.05). The three *MLL/KMT2A*-rearranged samples (AML002, AML004, AML086) that grouped together with the large NK cluster (Figure 1 and Figure 2) were excluded from the survival analysis. We used a hierarchical clustering approach to split patients into groups based on DNA methylation patterns of the 50 top CpG sites (Figure 4, Appendix A) and investigated clinical outcomes in each group of patients. In a second stage, we included all of the previously undefined diagnostic samples predicted by the DNA methylation classifier to each respective group, leading to the following sample sizes: t(8;21)/*RUNX1-RUNX1T1* (N = 19), *MLL/KMT2A*-rearranged (N = 24), and NK (N = 40), and repeated the survival analyses (Figure 4).

We identified variability in DNA methylation in the within-subtype groups of patients determined by diagnostic karyotyping and obtained two clusters of high and low methylation levels (Figure 4, heatmaps). Survival analysis on the groups of patients with subtypes known at AML diagnosis (Figure 4a,c,e right) revealed differences for all three subtypes (logrank *p*-value < 0.5) in RFS and OS while comparing the low and high methylation groups. Similar results were observed when the predicted undefined diagnostic samples were added in their corresponding subtype groups. Overall, the survival analysis revealed a relationship between patient outcome (relapse or death) and high/low methylation levels, in each of the three subtype groups (Figure 4). We included the DNA methylation group, treatment protocol, and risk group in a Cox proportional hazards regression analysis for RFS and OS. Treatment protocol had no systematic significant impact on outcome for NK or *MLL/KMT2A*-rearranged patients (Appendix A). The impact on risk grouping could only be evaluated in the patients with *MLL/KMT2A*-rearranged subtype treated on the NOPHO-2004 protocol, also showed no difference between standard and high risk (Appendix A). The t(8;21)*RUNX1-RUNX1T1* group was not analyzed in the Cox regression because the majority (16/19) of patients were treated on the NOPHO-2004 protocol and all were treated as standard risk.

## 4. Discussion

Herein, we investigated CpG site methylation in a relatively large cohort of Nordic pediatric AML patients and integrated the epigenomic data with clinical outcome. Overall, we found a large degree of DNA methylation variability between and within cytogenetic subtypes, which was informative for subtype determination as well as for identifying subsets of patients with inferior RFS and OS.

Pediatric AML represents many molecularly diverse entities. A large proportion (30%) of the samples in our cohort had undefined cytogenetic subtype, which is similar to the proportion of not otherwise specified (NOS) karyotype expected using the current classification system [2]. Our unsupervised clustering and subtype prediction modeling approaches suggest that several (N = 23) of these patients likely do belong to one of the recurrent subtypes (class probability > 80%) and that based on their DNA methylation patterns, these patients follow similar clinical paths as the subtype-confirmed cases. Although the DNA methylation classification provides a compelling case for re-annotating the subtypes of these patients, subtype-like cases in pediatric ALL [37,38] as well as adult t(8;21)-AML [8] without evidence of underlying genomic aberrations raises the possibility that the DNA methylation signatures are capturing another genetic event affecting similar pathways. Further molecular analysis of these cases will be needed in future studies to confirm their true subtype membership.

Increasing and compelling published evidence across the different hematological malignancies [39], adult [40,41] and pediatric AML [22], and the data presented herein support the notion that DNA methylation signatures at diagnosis are predictive of clinical outcome. Genes highlighted here in, such as *PRDM16* [42,43], *PER3* [44], *NOM1* [45] *BAHCC1* [46], *ZNF423* [47], *SETD7* [48], *RPTOR* [49], and others have been implicated in hematological disease development, progression or clinical outcome. Our results further strengthen two recent studies in pediatric AML focusing on genome-wide DNA methylation. First, in a similarly sized multicenter cohort (AML02) also using 450k arrays, DNA methylation patterns were associated with clinical outcome [22] and cytogenetic subtypes [17]. Reassuringly, several genes were identified in both studies, such as *CRIP2*, *LAMB2*, *USP2*. Second, a DNA hypomethylation signature associated with relapse was described in a small cohort of Italian pediatric t(8;21)/*RUNX1-RUNX1T1* patients [21]. Our data (Figure 4a,b) support and replicate this finding. Amongst the 18 t(8;21)/*RUNX1-RUNX1T1* cases presented herein, all but one patient who was in the group of patients with lower DNA methylation experienced a relapse. Further strengthening this finding is that the analytical and computational methods for DNA methylation analysis applied herein differ from those used by Zampini et al. [21]. The t(8;21)/*RUNX1-RUNX1T1* signature in particular, may reflect the presence of a CpG island methylator phenotype (CIMP), which has been associated with outcome in adult AML [41].

Although AMLs are generally known to incur fewer somatic mutations than other cancer types, several genes are well established to be recurrently mutated in AML and may hold prognostic information [34,50]. *WT1* and *FLT3-ITD* mutations are markers of poor outcome in pediatric AML, while *NPM1* and *CEBPA* mutations may have a favorable outcome. We found that our NK-AML subgroup was split into two distinctive clusters based on DNA methylation, and that mutations in *WT1*, *FLT3*, and *CEBPA* were overrepresented in one of the clusters (Cluster B), while the other cluster contained few mutations (Cluster A). Few (N = 3) *NPM1* mutations were observed in our dataset, two in Cluster B and one in Cluster A. We observed a clearly defined sub-cluster within Cluster B, containing all of the patients with *CEBPA* mutations, a pattern which has been reproduced in adult AML cohorts [8,11]. However, the limited resolution of the four gene panel applied herein (*CEBPA*, *FLT3*, *NPM1*, and *WP1*) in addition to incomplete data for 27% of the patients, may give a partial view of the mutational status and furthermore may have missed key molecular changes associated with the two DNA-methylation defined NK clusters. Future studies are warranted to more thoroughly investigate if there are any other genetic differences that be driven by other recurrent mutation(s) or cryptic translocations that disrupt pathways resulting in altered DNA methylation landscapes. Other biological differences, perhaps by variation in DNA methylation by age of the patient, may also contribute to these findings. Another limitation of the present study is the unavailability of MRD data for the patients analyzed herein. Inclusion of patients with MRD data available would be important information to analyze in future studies to investigate if DNA methylation grouping can provide additional prognostic information.

Our study not only adds to the growing wealth of knowledge about clinical use of epigenetic signatures, but also highlights the possibility that exclusive use of cytogenetic markers at diagnosis may underestimate the variability within and between classical subtypes. The randomized clinical trials in recent years have struggled to improve outcomes for pediatric AML patients, and this failure may be in part due to applying blanket treatment strategies across AML subtypes, which may not prove effective for all patients [14]. The importance of cataloguing epigenetic changes within genetic subgroups of pediatric AML is underscored given new possible alternatives for future treatment of acute leukemias with DNA methylating agents such as Temozolomide, demethylating agents, and histone deacetylase inhibitors (HDACi). There are several ongoing studies to explore their efficacy in the treatment of hematological cancers [51]. The demethylating agents azacytidine and decitabine have been approved for treatment of myelodysplastic syndrome and a combination of demethylating agents and HDACi has shown promising results in AML patients [52,53].

## 5. Conclusions

In summary, we show that DNA methylation is associated with cytogenetic subtype and clinical outcome in pediatric AML. Further analyses of DNA methylation levels in t(8;21)/*RUNX1-RUNX1T1* and NK subtypes, in particular, are warranted to investigate the underlying molecular and cellular pathways leading to the intra-subtype DNA methylation variability observed in this cohort and to evaluate if DNA methylation signatures hold prognostic information in larger cohorts.

## Figures and Tables

**Figure 1 genes-12-00895-f001:**
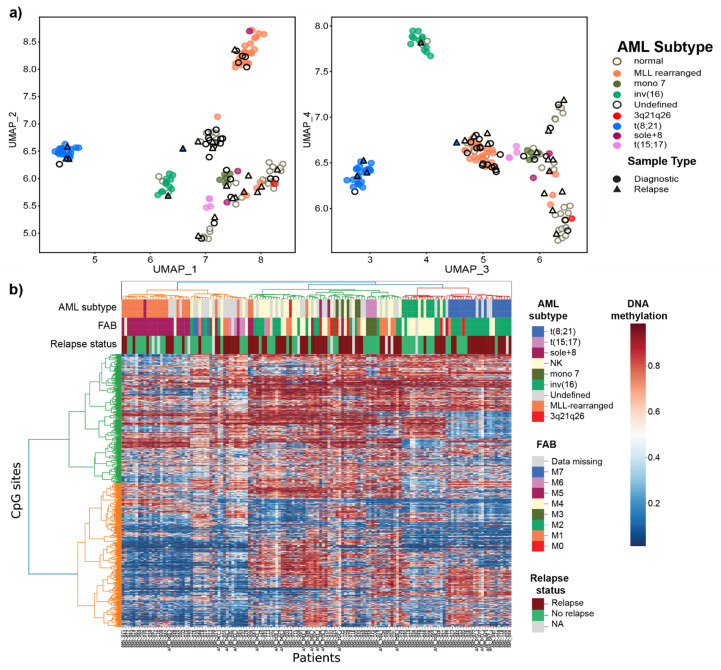
Data representation in space for the 142 pediatric AML patient samples for the selected 1300 CpG sites. (**a**) DNA methylation data projection in 2D space in a UMAP plot for UMAPs 1–2 (left) and 3–4 (right). Each point represents a patient sample. All samples are labeled by cytogenetic subtype, samples with unknown subtype (undefined) are shown as black circles with no fill (N = 24, diagnostic samples), and samples taken at relapse are shown by filled triangles colored by cytogenetic subtype (N = 19, relapse samples). (**b**) DNA methylation heatmap ordered by hierarchical clustering. Each row in the heatmap denotes a CpG site and each column is a patient. Cytogenetic subtype, FAB classification, and relapse status are shown as annotated bars over the plot.

**Figure 2 genes-12-00895-f002:**
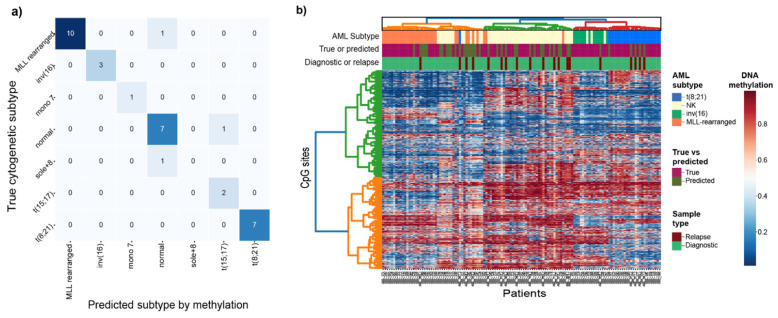
(**a**) Confusion matrix for the cytogenetically defined (diagnostic-known) vs. DNA methylation predicted cytogenetic subtypes (N = 33). (**b**) Heatmap and hierarchical clustering of 127 samples taken at AML diagnosis or relapse from patients belonging to the four subtypes most common subtypes; t(8;21)/*RUNX1-RUNX1T1*, inv(16)/*CBFB-MYH11*, *MLL/KMT2A*-rearranged and NK. The methylation status of the selected 1300 CpG sites are plotted in the heatmap. The samples are in columns labeled by AML subtype, whether the sample was cytogenetically defined (true/diagnostic-known) or cytogenetically undefined (DNA methylation predicted), and if the sample was taken at AML diagnosis or relapse.

**Figure 3 genes-12-00895-f003:**
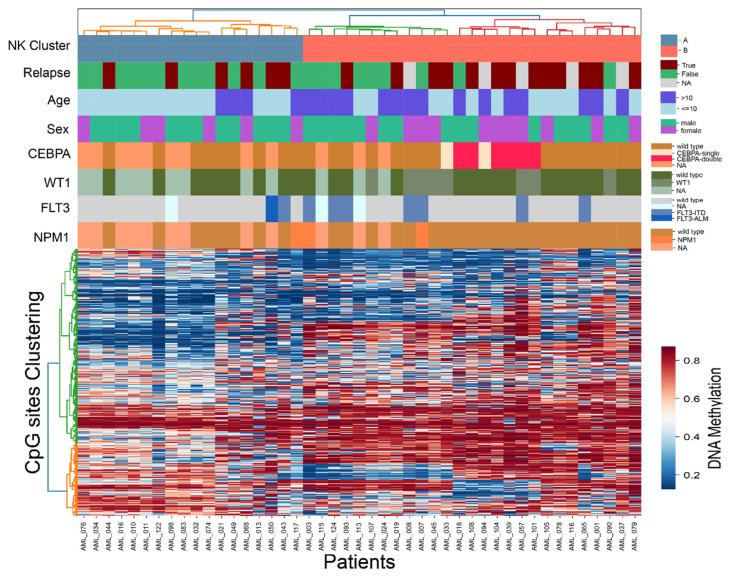
Heatmap and hierarchical clustering of the 45 NK samples across 1300 CpG sites, including 30 cytogenetically defined (diagnostic-known) patients and 15 DNA methylation predicted NK patients. Each row in the heatmap denotes a CpG site and each column is a patient. The labels at the top of the heatmap represent clinical and molecular features of interest, including the Cluster denotation A and B.

**Figure 4 genes-12-00895-f004:**
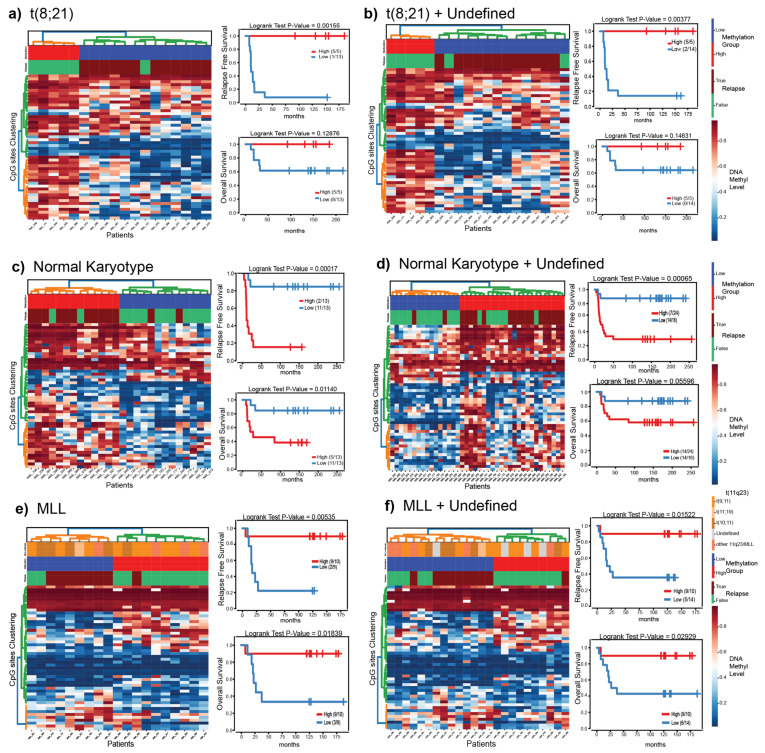
Survival analysis for t(8;21)/*RUNX1-RUNX1T1* (**a**,**b**), NK (**c**,**d**), and *MLL/KMT2A*-rearranged (**e**,**f**) patient samples based on the 50 most significant CpGs that separate the diagnostic samples of patients who later went on to relapse from those who did not. Panels (**a**,**c**,**e**) contain patients with a confirmed diagnostic subtype (diagnostic-known). Panels (**b**,**d**,**f**) contain diagnostic-known in addition DNA methylation predicted samples from the undefined group. In each panel, the heatmap (left), is ordered by hierarchical clustering. Samples along the *x*-axis are split into two groups color coded by high (red) or low (blue) overall methylation level. The numbers in the legend represent the fraction of patients who did not experience an event. Kaplan–Meier curve for relapse free survival analysis (upper right) and overall survival (lower right) of the two groups identified by clustering. The *x*-axis represents the time until event (months) and events are plotted on the *y*-axis.

**Table 1 genes-12-00895-t001:** Cytogenetic subtype, age range, central nervous system (CNS) involvement, stem cell transplantation (SCT), and FAB classification for the 142 pediatric AML patient samples. The number of samples taken at AML diagnosis and relapse are presented. When known, the subtype of the relapse samples is indicated as the same subtype as their diagnostic pair.

Cytogenetic Subtype	Normal Karyotype (NK)	*MLL/KMT2A* Rearranged	Undefined	t(8;21) *RUNX1/RUNX1T1*	inv(16) *CBFB/MYH11*	mono 7	t(15;17) *PML-RARA*	sole +8	3q21q26
Number of diagnostic Samples	30	25	24	19	12	5	4	3	1
Number of relapse samples	8	1	5	4	1	-	-	-	-
Age range	1–18	0–16	0–17	2–16	1–17	1–5	3–16	3–14	14
CNS involvement	-	-	-	-	-	-	-	-	-
Yes	4	1	2	6	-	-	-	-	-
No	34	25	27	17	12	4	3	3	1
Missing	-	-	-	-	1	1	1	-	-
SCT	-	-	-	-	-	-	-	-	-
Yes	8	3	1	-	-	5	-	2	-
No	30	23	27	23	13	-	3	1	1
Missing	-	-	1	-	-	-	1	-	-
FAB	-	-	-	-	-	-	-	-	-
M0	2	-	-	1	-	2	-	1	-
M1	7	-	3	1	-	1	-	1	1
M2	14	1	3	19	3	1	-	-	-
M3	1	-	1	-	-	-	4	-	-
M4	10	1	10	2	9	-	-	-	-
M5	-	24	4	-	-	1	-	1	-
M6	3	-	2	-	-	-	-	-	-
M7	-	-	4	-	-	-	-	-	-
Missing	1	-	2	-	1	-	-	-	-

**Table 2 genes-12-00895-t002:** Precision, recall, F1 scores per subtype, as well as overall accuracy of the classifier for the four subtypes with >5 patients in the group. High precision scores indicate the low number of false positives (FP) while high recall scores the low number of false negatives (FN).

	Precision	Recall	F1 Score	Total samples Test Set	Total Samples Train Set
*MLL/KMT2A*-rearranged	1	0.91	0.95	11	14
inv(16)/*CBFB-MYH11*	1	1	1	3	9
Normal Karyotype (NK)	0.78	0.88	0.82	8	22
t(8;21)/*RUNX1-RUNX1T1*	1	1	1	7	12

**Table 3 genes-12-00895-t003:** Top selected CpGs and genes per subtype based on a one-vs.-rest CpG Site Selection with adjusted *p*-value threshold 0.05.

Subtype	N CpG Sites (Adjusted *p* Value < 0.05)	N CpG Sites Unique to Subtype (N Genes)	Gene Names (CpG IDs) Unique to Subtype
Normal Karyotype	569	6 (5)	*ZNF793* (cg15139588), *APBA2* (cg15605858), *PRDM16* (cg02390319), *PPP1R14A* (cg02571816), *ACCN1* (cg03745383)
*MLL/KMT2A*-rearranged	873	59 (33)	*KIAA1755* (cg14003035), *PLAUR* (cg27340480), *PER3* (cg05803631), *ASB2* (cg09341793), *KLK4* (cg26827876), *BARHL2* (cg18322569), *L1TD1* (cg23049458), *ARPC1B* (cg10428938), *ST8SIA6* (cg17256364), *NKX6-2* (cg11174855), *WNT5A* (cg19554389), *HOXA5* (cg12128839, cg25307665), *NFIX* (cg06744585), *SNED1* (cg25241559, cg09991306), *TNXB* (cg12694372, cg10923662, cg16834823, cg01992382), *MSX2* (cg06013117), *MAPK8IP1* (cg08214808), *BNIP3* (cg18477674), *CASR* (cg19108881), *HECW1* (cg24384918), *PCDHA1; PCDHA2; PCDHA3; PCDHA4; PCDHA5; PCDHA6; PCDHA7; PCDHA8* (cg19596110), *PITX1* (cg00396667), *KCNN1* (cg07857792), *TMEM132D* (cg20168964), *NPSR1* (cg20276677), *LOC732275* (cg16709904), *NOM1* (cg02413092), *SPEG* (cg16440561), *EDARADD* (cg09164898), *THBS4* (cg26286839), *HOOK2* (cg06417478), *LOC254559* (cg09969277), *DCC* (cg25204852)
mono7	330	24 (12)	*BCL2* (cg25059899), *ZNF577* (cg03562414, cg24794228, cg11269599, cg10635122), *ZNF154* (cg21790626, cg27049766, cg26465391), *ARRB2* (cg07971820, cg02286380), *SKI* (cg25139649), *ERCC3* (cg06373940), *RPTOR* (cg09929238), *FBXO47* (cg04120272), *DLL1* (cg00084338), *C1orf86; LOC100128003* (cg26227225), *CYP1A1* (cg22549041), *PLD6* (cg24578857, cg19093370)
inv(16)/*CBFB-MYH11*	571	22 (15)	*DPF3* (cg13588403), *IFLTD1* (cg13134916), *BAHCC1* (cg06636541), *LEPR* (cg16987305), *AFAP1* (cg22079161), *PRHOXNB* (cg20101529), *ANK1* (cg19537719), *SHISA6* (cg13330559), *PRDM16* (cg03337482), *MUC4* (cg05834845), *C22orf34* (cg20744362), *LY96* (cg23732024), *ZNF423* (cg26929700, cg04086531), *GNG7* (cg26988138), *CNTD2* (cg08871608)
t(8;21)/*RUNX1-RUNX1T1*	723	27 (15)	*SMTNL2* (cg13375589), *TUSC1* (cg13811417), *PDLIM3* (cg14632696), *CYP27C1* (cg08022717), *PCDHA1; PCDHA10; PCDHA11; PCDHA2; PCDHA3; PCDHA4; PCDHA5; PCDHA6; PCDHA7; PCDHA8; PCDHA9* (cg26514430), *RYR2* (cg07790615), *TACSTD2* (cg13443627), *FBXL7* (cg26134895), *VSTM2A* (cg19868631), *MARCH11* (cg25092681, cg01791874,cg00339556, cg16150752, cg17712694), *IGSF21* (cg15564444), *TTBK1* (cg16620382), *SHROOM1* (cg21811204), *ELOVL4* (cg04107099), *SETDB1* (cg15448220)
t(15;17)/*PML-RARA*	328	8 (8)	*NFYC* (cg16167741), *C7orf50* (cg23657099), *IGDCC4* (cg00776960), *SETD7* (cg02409722), *SCHIP1* (cg23553912), *SYNE1* (cg02796568), *WBSCR17* (cg02300154), *C21orf7* (cg08854834)

## Data Availability

DNA methylation data are available from the authors upon request via: 10.17044/scilifelab.14666127.

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
