# Peer review of "DNA Methylation Signatures Predict Cytogenetic Subtype and Outcome in Pediatric Acute Myeloid Leukemia (AML)"

_genes, 2021, doi:10.3390/genes12060895_

Round 1

Reviewer 1 Report

Krali et al addresses whether DNA methylation signatures could predict cytogenetic subtypes and prognosis in pediatric AML. Even though similar studies have been reported by others, in this study the data presented are of interest.

Specific points:

Figure 1: In general, more explanation for this figure is needed in the main text. The results of Figure 1 should be elaborated.

-The author should clarify what “the remaining data” means. How many CpG sites have they chosen for Figure 1?

-The relapse samples should be colour-coded based on their cytogenetic subtype to show whether they cluster with their cytogenetic subtype at diagnosis.

- NK samples do not cluster together. The authors should elaborate on this.

Figure S4 should be in the methods section.

Figure S3 is mentioned after Figure S6. Sup. Figures should be numbered in the order they appear in the text.

Figure 2b: The authors should comment on whether the genetic mutations might contribute to the deviation seen in the Figure, especially for the NK subtype. Have they checked the mutation status of these samples?

In Line 269, “Survival analysis on the groups of patients with 268 subtypes known at AML diagnosis (Figure 3 a, c, d right)” Figures should be 3 a, c, e for the known subtype at diagnosis.

Figure 3: When the predicted undefined diagnostic samples were added in MLL subtype groups, there is no difference in OS between low and high methylation groups. Could the prediction of the undefined samples not be accurate for the MLL subtype? Did the authors test this correlation using published data (e.g. Koldopbskiy, et al.)?

Figure S7: The results are not strong enough for this section to show any relationship with mutational status and NK clustering, prognosis. It is unclear why the authors focus on only four mutations, CEBPA, WT1, FLT3 and NPM1. Group A is WT for these mutations, but perhaps there is another mutation that plays a major role to define cluster A. Moreover, it is known that the mutational status of CEBPA and FLT3 are associated with different outcomes, CEBPA with favourable and FLT3 with poor prognosis. However, the authors analyse the presence of these mutations together in cluster A and B. It will be more informative, if the authors show the mutational status of each sample in Figure S7. Finally, the mutational status is not available for all the samples, which may interfere with the analysis.

There is not enough evidence for the conclusion: “that the presence of CEBPA, FLT3, WP1, and possibly NPM1 mutations may be correlated with the differential DNA methylation signature associated with poor outcome in NK-AML”.

Author Response

Reviewer 1

Comments and Suggestions for Authors

Krali et al addresses whether DNA methylation signatures could predict cytogenetic subtypes and prognosis in pediatric AML. Even though similar studies have been reported by others, in this study the data presented are of interest.

Specific points:

Figure 1: In general, more explanation for this figure is needed in the main text. The results of Figure 1 should be elaborated.

We added additional description and explanation of the results presented in Figure 1 in the text on Lines 171-180 of the revised version.  

-The author should clarify what “the remaining data” means. 

We clarified this point on line 171 of the revised version that the “remaining data” is the 1300 CpG sites that were retained after the CpG site selection procedure. 

How many CpG sites have they chosen for Figure 1? 

In the legend of Figure 1 we state that 1300 CpG sites are used. See also response to the comment above, which also clarifies the number of CpG sites plotted in Figure 1 in the text. 

-The relapse samples should be colour-coded based on their cytogenetic subtype to show whether they cluster with their cytogenetic subtype at diagnosis.

We agree with the reviewer that color-coding the relapse samples by cytogenetic subtype improves the interpretability of this figure. We have therefore changed the coloring of the relapse samples in the revised version of the manuscript as suggested. 

- NK samples do not cluster together. The authors should elaborate on this.

We moved the section (previously 3.5) about the mutational analysis in the NK group to section 3.3 in the new version, specifically to elaborate on the split in the NK cluster. In addition, we made several changes to this section in response to subsequent suggestions from the reviewer regarding the mutational analysis, see below.

Figure S4 should be in the methods section.

We changed the order of the Supplemental Figures and Figure S4 is now Figure S1 and referred to in the methods section in the revised version of the manuscript. 

Figure S3 is mentioned after Figure S6. Sup. Figures should be numbered in the order they appear in the text.

 We have corrected this error in the revised version of the manuscript and now the Supplemental Figures and Tables are ordered by how they appear in the manuscript. 

Figure 2b: The authors should comment on whether the genetic mutations might contribute to the deviation seen in the Figure, especially for the NK subtype. Have they checked the mutation status of these samples?

Based on this comment we moved section 3.5 describing the mutational analysis in NK to earlier in the manuscript (now Section 3.3) and revised this section based on the other comments raised by the reviewer in regards to the mutational analyses performed herein. 

In Line 269, “Survival analysis on the groups of patients with 268 subtypes known at AML diagnosis (Figure 3 a, c, d right)” Figures should be 3 a, c, e for the known subtype at diagnosis.

 We have corrected this in the revised version of the manuscript.

Figure 3: When the predicted undefined diagnostic samples were added in MLL subtype groups, there is no difference in OS between low and high methylation groups. Could the prediction of the undefined samples not be accurate for the MLL subtype? Did the authors test this correlation using published data (e.g. Koldopbskiy, et al.)?

The predicted class probability as defined by our classifier is very high (>98%) for all of the predicted MLL-rearranged samples, and not different from the other larger subtypes, which speaks against the fact that the MLL classifier is more inaccurate than for the other subtypes. 

We are, however, grateful that the reviewer pointed this out. In the original version of the manuscript, we noted that the hierarchical clustering in Figure 3f was driven mainly by three MLL-rearranged samples, which have distinctively different DNA methylation profiles than the rest of the group. These samples also clustered independently in the previous version Figure 3e, but they did not drive their own split in the first branch of the cluster. When more data were added (i.e. the predicted diagnostic MLL samples), the three “outlier” samples branched early in the clustering and drove the cluster definition. Our method, which splits the groups into two, resulted in one group containing only the three outliers in original Figure 3f. 

The three outlier samples include three known MLL-rearranged samples: AML_002/t(11;19), AML_004/t(11;19), and AML_086/t(9;11). Upon closer examination of these three samples, we noted that not only do they cluster differently in Figure 3f, but that they cluster with the large cluster of NK samples in the UMAP plot (Figure 1a) and in the hierarchical clustering (Figure 1b). Our DNA methylation classifier confirmed that AML_002 and AML_004 are indeed MLL-rearranged. However, our classifier suggests that AML_086 appears to be NK based on its methylation profile. We therefore decided to go back and repeat the MLL-rearranged analysis presented in Figure 3e-f excluding the three outliers. In the revised version is now Figure 4. The end result in Figure 4e is the same, we observe two clusters with large deviation in RFS/OS in the diagnostic MLL-rearranged samples (excluding the three outliers). The biggest improvement is in Figure 4f, where we now have two more defined, separated clusters with where the observed difference in RFS/OS is retained. 

Figure 4 is now updated in the revised version of the manuscript to reflect the aforementioned changes. In addition, we have explained why we removed the three outlier MLL samples (AML_002, AML_004, AML_086) from the survival analysis (see section 3.5, specifically lines 304-306)

We compared all of our gene lists (Table 3 and Supplementary Tables S5-S6) for MLL-rearranged as well as the other cytogenetic subtypes to previous studies on AML DNA methylation to provide added support for our signatures. As noted on lines 289-292 of the results and 358-363 of the discussion, we identified several genes overlapping previous studies, which is encouraging especially given our approach to select a small number of CpGs (N=1300). To the best of our knowledge most of the prior studies on DNA methylation in pediatric AML have not made per-patient individual DNA methylation values available in repositories, which limits our ability widely replicate or test correlations using other published data. The data is indeed available for Koldobskiy et al (9 MLL-rearranged cases by WGBS), however the data available online is based on mean DNA methylation levels of 150bp bins tiled throughout the genome. Given that the Koldobskiy dataset is limited to MLL and because base pair resolution methylation values were not readily available, for clarity and cohesiveness of our manuscript, we chose to omit this type of validation/comparison.

Figure S7: The results are not strong enough for this section to show any relationship with mutational status and NK clustering, prognosis. It is unclear why the authors focus on only four mutations, CEBPA, WT1, FLT3 and NPM1. Group A is WT for these mutations, but perhaps there is another mutation that plays a major role to define cluster A. Moreover, it is known that the mutational status of CEBPA and FLT3 are associated with different outcomes, CEBPA with favourable and FLT3 with poor prognosis. However, the authors analyse the presence of these mutations together in cluster A and B. It will be more informative, if the authors show the mutational status of each sample in Figure S7. Finally, the mutational status is not available for all the samples, which may interfere with the analysis.

There is not enough evidence for the conclusion: “that the presence of CEBPA, FLT3, WP1, and possibly NPM1 mutations may be correlated with the differential DNA methylation signature associated with poor outcome in NK-AML”.

We focused on the mutational status of four specific genes (CEPBPA, WT1, FLT3 and NPM1) because mutations in these genes were screened for at AML diagnosis according to best practice procedures for clinical workup at the time this patient cohort was collected (1997-2008). In total we had 45 known and predicted NK-AML patients (15 are predicted and 30 of known NK subtype). Of these, 33 patients were screened for all four mutations, nine patients screened for FLT3 only, and 3 patients were not screened at all. As the reviewer suggested, we now plot the mutations in each sample as rows in about the heatmap in Figure 3 (previously supplementary figure S7). 

We agree with the reviewer’s comment regarding our analysis and have revised the manuscript to improve the results and discussion about the two NK groups identified based on DNA methylation herein in the following ways:

  1. We moved up the mutational analysis (new section 3.3 in the revised version) to improve the flow of the manuscript.
  2. Instead of focusing only on mutational status in section 3.3, we analyzed all clinical data available for the NK patients (age, sex, FAB, mutational status, etc) in order to investigate possible explanations for the two clusters. We found that Cluster B has significantly higher frequency of any of these four mutations (fisher’s exact p-value 0.0002) and are significantly older (t-test p-value 0.00004).This is now discussed in section 3.3 of the revised version of the manuscript (lines 256-274), in the discussion (see comment 3, below) and illustrated in Figure 3.
  3. We removed the statement that the presence of “CEBPA, FLT3, WP1 and possibly NPM1 mutations were associated with the poor outcome group.” We further elaborate in the discussion that although our dataset is small and incomplete for mutations genome-wide, we did note two differences between the two NK clusters. First, Cluster B has a higher frequency of mutations in these four genes. In particular, we see one sub-cluster containing all of the patients with CEBPA mutations, as has been previously observed in adult AML (Figueroa et al, 2010; Gebhard et al, 2019). Second, we find a difference in age of the patients in the two clusters, where the average age of cluster A is 4.5 years and cluster B is 10.8 years. But we clarify that further studies would be needed to see if there is any other genetic difference driving the two DNA methylation clusters in the NK group, which may be driven in part by other biological differences in the age of the patients, or perhaps by underlying variation in DNA methylation by age (lines 384-395).

Reviewer 2 Report

In this study, the authors have performed the genome-wide DNA methylation analysis on a large group of 125 children with AML and then they have investigated whether variable DNA methylation patterns hold any diagnostic or prognostic information in this leukemia.

The actual survival rates in pediatric AML are still unsatisfactory, so its treatment still needs further improvement. Most currently used prognostic classification systems consider cytogenetic and molecular factors and disease response by MRD with host factors and clinical characteristics for determining risk. Future risk stratification is likely to include additional molecular/genomic factors as well as epigenetic factors and drug sensitivity testing. Therefore, I believe that Krali et al. study is important and, together with the results obtained by others, may contribute to a better stratification of pediatric AML patients in the future. These data will also give hope for the incorporation of molecularly targeted therapeutics into frontline treatment for improving survival while decreasing treatment-related toxicity.

In my opinion, this manuscript is well written and the methodology used is correct. While I have no comments about the results from points 3.1-3.3, linking the methylation profile with the cytogenetic subtype of AML, the part related to the methylation patterns impact on the outcome needs to be revised.

Major issues

  1. The OS and RFS have been calculated together for patients treated according to two different protocols (NOPHO93 and NOPHO2004)

2 The outcome analysis in AML should also include the evaluation of response to treatment like MRD level after induction chemotherapy completion. Otherwise, the assessment of the DNA methylation profile as a prognostic factor may be skewed because of the significant impact of the treatment’s related mortalities on overall survival.

  1. To minimalize the different treatment’s impact on the outcome I also suggest the additional calculating of OS/RFS for standard and high risk group separately.

There are also some minor issues:

  1. More clinical data (e.g. range of age, FAB, CNS status, MRD level after induction, risk group, SCT etc) of the patients should be provided
  2. lines160 and 178 –as the table header says, the FAB classification should be included in table 1.
  3. Table 2. The subtypes: mono 7, sole+8, t(15;17)/PML-RARA are too small for precision, recall and F1 scores calculating, so they should be excluded
  4. Figure 3. The diagrams with Kaplan-Meier’s curves are in very low resolution and thus scarcely legible. Furthermore, each curve could be labeled with “n= the number of patients”

Author Response

Reviewer 2

Comments and Suggestions for Authors

In this study, the authors have performed the genome-wide DNA methylation analysis on a large group of 125 children with AML and then they have investigated whether variable DNA methylation patterns hold any diagnostic or prognostic information in this leukemia.

The actual survival rates in pediatric AML are still unsatisfactory, so its treatment still needs further improvement. Most currently used prognostic classification systems consider cytogenetic and molecular factors and disease response by MRD with host factors and clinical characteristics for determining risk. Future risk stratification is likely to include additional molecular/genomic factors as well as epigenetic factors and drug sensitivity testing. Therefore, I believe that Krali et al. study is important and, together with the results obtained by others, may contribute to a better stratification of pediatric AML patients in the future. These data will also give hope for the incorporation of molecularly targeted therapeutics into frontline treatment for improving survival while decreasing treatment-related toxicity.

In my opinion, this manuscript is well written and the methodology used is correct. While I have no comments about the results from points 3.1-3.3, linking the methylation profile with the cytogenetic subtype of AML, the part related to the methylation patterns impact on the outcome needs to be revised.

Major issues

  1. The OS and RFS have been calculated together for patients treated according to two different protocols (NOPHO93 and NOPHO2004).

Correct, we analyzed the patients treated on the NOPHO93 and NOPHO2004 protocols together. We did this because the two protocols were very similar. Both had response guided timing of the second course and identical four course consolidation. The only differences were a switch from doxorubicin to idarubicin in course one and that all received the course “AM” (cytarabine and mitoxantrone) in NOPHO2004 as second course which was only given to poor responders in NOPHO93.

To support this, we now include a Cox Regression analysis with protocol as a covariate in the revised version of Supplementary Figure S7-9 and in the main text on lines 321-329 of the revised version, which demonstrates that outcome of the DNA methylation groups are independent of protocol for MLL-rearranged and NK samples. The only exception for the NK OS when adding the undefined samples (Supplemental Figure S7b) which was marginally significant p= 0.05. The patient numbers are small, so this analysis cannot be considered as definitive, but it demonstrates that treatment protocol is not driving the outcome. For the t(8;21) subtype, we had 19 patients in total (18 known + 1 predicted) and only three were treated on the NOPHO93 protocol. This low number prohibits comparison between the NOPHO-93 and NOPHO2004 protocols in this group.

  1. The outcome analysis in AML should also include the evaluation of response to treatment like MRD level after induction chemotherapy completion. Otherwise, the assessment of the DNA methylation profile as a prognostic factor may be skewed because of the significant impact of the treatment’s related mortalities on overall survival.

We thank the reviewer for this suggestion, however MRD data are not available for the majority of the patients included in this study, which reflects the time period of diagnosis (1997-2008). We agree that in order to definitely show a DNA methylation profile as an independent prognostic factor, in future studies inclusion of patients with MRD data would be important.  We now discuss this limitation in the discussion on lines 396-399

  1. To minimalize the different treatment’s impact on the outcome I also suggest the additional calculating of OS/RFS for standard and high risk group separately.

We only have risk grouping for the NOPHO2004 protocol. Risk grouping (standard/high) wasn’t performed in NOPHO93.

For MLL-rearranged on the NOPHO2004 protocol, we had 16 patients (13 known MLL and 3 predicted). 10 were SR and 6 were HR. Although the number is low, we were able to perform a cox regression analysis for this subtypes and it is presented now in Supplemental Figure S9. 

For the other subgroups t(8;21)/NK treated on NOPHO2004, all t(8;21) patients were standard risk of the 14 NK patients (10 known + 4 predicted) only 3 patients on were in the High Risk group. We therefore could not calculate hazard with so few patients for these two groups. 

There are also some minor issues:

  1. More clinical data (e.g. range of age, FAB, CNS status, MRD level after induction, risk group, SCT etc) of the patients should be provided

Table 1 now includes age range, CNS status, SCT and FAB. In Supplementary Table S1 we also added age, treatment protocol, risk group (for NOPHO2004 protocol).  As MRD data was not available for most of the patients, we were unfortunately not able to include it in the tables. 

  1. lines160 and 178 –as the table header says, the FAB classification should be included in table 1.

We apologize for this error. In the revised version of the manuscript we now show FAB, age range, see comment #1 above.

  1. Table 2. The subtypes: mono 7, sole+8, t(15;17)/PML-RARA are too small for precision, recall and F1 scores calculating, so they should be excluded

We agree with the reviewer that the small subtypes are too small for precision recall and F1 scores, we have removed them from Table 2. 

  1. Figure 3. The diagrams with Kaplan-Meier’s curves are in very low resolution and thus scarcely legible. Furthermore, each curve could be labeled with “n= the number of patients”

We have revised this figure so it should now be of higher quality and we made sure the number of patients in each group is more legible. In addition to the embedded figure in the word document, we also submitted all figures in high resolution JPEG format. 

Round 2

Reviewer 1 Report

Following the revision to the article, the authors have addressed major comments and have sufficiently improved their paper.